# Comprehensive Leaf Cell Wall Analysis Using Carbohydrate Microarrays Reveals Polysaccharide-Level Variation between *Vitis* Species with Differing Resistance to Downy Mildew

**DOI:** 10.3390/polym13091379

**Published:** 2021-04-23

**Authors:** Yu Gao, Xiangjing Yin, Haoyu Jiang, Jeanett Hansen, Bodil Jørgensen, John P. Moore, Peining Fu, Wei Wu, Bohan Yang, Wenxiu Ye, Shiren Song, Jiang Lu

**Affiliations:** 1Center for Viticulture and Enology, Department of Plant Science, School of Agriculture and Biology, Shanghai Jiao Tong University, Shanghai 200240, China; yugao@sjtu.edu.cn (Y.G.); jianghaoyu@sjtu.edu.cn (H.J.); fupeining@sjtu.edu.cn (P.F.); wuwei_hello@sjtu.edu.cn (W.W.); ybh622@sjtu.edu.cn (B.Y.); yewenxiu@sjtu.edu.cn (W.Y.); sr.song@sjtu.edu.cn (S.S.); 2Forestry and Pomology Research Institute, Shanghai Academy of Agricultural Sciences, Shanghai 201403, China; yinxiangjingsmile@163.com; 3Shanghai Key Lab of Protected Horticultural Technology, Shanghai Academy of Agricultural Sciences, Shanghai 201403, China; 4Department of Plant and Environmental Sciences, Faculty of Sciences, University of Copenhagen, 1871 Frederiksberg C, Denmark; jeha@plen.ku.dk (J.H.); boj@plen.ku.dk (B.J.); 5Department of Viticulture and Oenology, South African Grape and Wine Research Institute, Stellenbosch University, Stellenbosch 7602, South Africa; moorejp@sun.ac.za

**Keywords:** cell wall, grapevine, oomycete, polysaccharide microarray, monoclonal antibodies

## Abstract

The cell wall acts as one of the first barriers of the plant against various biotic stressors. Previous studies have shown that alterations in wall polysaccharides may influence crop disease resistance. In the grapevine family, several native species (e.g., Chinese wild grapevine) show a naturally higher resistance to microbial pathogens than cultivated species (e.g., *Vitis vinifera*), and this trait could be inherited through breeding. Despite the importance of the cell wall in plant immunity, there are currently no comprehensive cell wall profiles of grapevine leaves displaying differing resistance phenotypes, due to the complex nature of the cell wall and the limitations of analytical techniques available. In this study, the cutting-edge comprehensive carbohydrate microarray technology was applied to profile uninfected leaves of the susceptible cultivar (*Vitis vinifera* cv. “Cabernet Sauvignon”), a resistant cultivar (*Vitis amurensis* cv. “Shuanghong”) and a hybrid offspring cross displaying moderate resistance. The microarray approach uses monoclonal antibodies, which recognize polysaccharides epitopes, and found that epitope abundances of highly esterified homogalacturonan (HG), xyloglucan (with XXXG motif), (galacto)(gluco)mannan and arabinogalactan protein (AGP) appeared to be positively correlated with the high resistance of *Vitis amurensis* cv. “Shuanghong” to mildew. The quantification work by gas chromatography did not reveal any significant differences for the monosaccharide constituents, suggesting that polysaccharide structural alterations may contribute more crucially to the resistance observed; this is again supported by the contact infrared spectroscopy of cell wall residues, revealing chemical functional group changes (e.g., esterification of pectin). The identification of certain wall polysaccharides that showed alterations could be further correlated with resistance to mildew. Data from the use of the hybrid material in this study have preliminarily suggested that these traits could be inherited and may be applied as potential structural biomarkers in future breeding work.

## 1. Introduction

The domesticated grapevine (*Vitis vinifera*) is susceptible to many pathogens, which may lead to huge losses in production. Unlike the vine trunk, which contains a higher percentage of cellulosic and lignin components that are more recalcitrant to biological degradation, the grapevine leaves with a higher primary cell wall content are markedly more vulnerable to opportunistic pathogenic microorganisms, e.g., fungi [1]. Necrotrophic pathogens can secrete a wide range of cell wall-degrading enzymes (CWDEs) to effect almost the complete deconstruction of plant tissues [2]. In contrast to necrotrophs, which acquire nutrients from dead tissue, biotrophs need the host to be alive during infection/feeding and are found to implement “milder” penetration procedures [3]. Studies of biotrophs suggest that some of the enzymes employed can also be secreted from the haustoria of biotrophic pathogens interfacing with plants for more effective invasion; these may include a set of pectinases and hemicellulases depending on their targeted tissue types [4,5]. In contrast, plants have also evolved various defense mechanisms at the structural and molecular levels under different types of stresses [6]. These not only elevate their chances of survival and continuity, but they also act as selection pressures acting on the Vitis species during plant evolution. The biodiversity of the Vitis genus provides researchers with many valuable plant genetic resources for the investigation of molecular defense mechanisms against fungi, as well as contributing towards breeding programmes. Several studies have analyzed the differences between crops with close genetic relatedness from a cell wall structural perspective, and revealed compositional changes in polysaccharides and proteins, which are probably adaptive [4,5,6,7,8,9]. In addition to the cell wall compositional and structural differences observed, recent studies have also shown that the host plant is found to secrete inhibiting proteins that de-activate CWEDs as a means of protection [10,11,12,13,14]. These studies have broadened our views on the roles of the cell wall in plant-pathogen interactions. However, there are very few comparative studies from a cell wall structural point of view in grapevine–pathogen interactions, even though a variation in defense responses have been observed in the Vitaceae and genus Vitis [15].

Grapevine downy mildew is recognized as one of the more serious diseases to infect *Vitis vinifera* and is caused by the oomycete *Plasmopara viticola*. It is found to penetrate the grapevine leaves through stomata, and then develop haustoria on the hyphae to unravel the structure of the middle lamella and primary cell wall for nutrient uptake and later reproduction [16]. Microscopy studies have shown that a number of species in the Vitis genus could slow down the growth of hyphae in the leaf tissue; these include some Chinese native species (e.g., *Vitis amurensis*) and a number of American species [15,17,18]. To date, studies on the identification of resistance genes using QTL localisation and other molecular biological procedures have resulted in valuable progress [19,20,21]. Furthermore, studies from a phenotypical and physiological perspective have also provided useful information, in turn, aided by technology development. For instance, callose (beta-1,3-glucan) deposition was clearly observed near the tip end of hyphae of resistant grapevine species by using electron microscopy [15]. This clearly indicated that grapevine leaf cell walls could undergo localised structural alterations to adapt to external stresses, such as infection. However, it is regrettable that we still lack information on the comprehensive cell wall composition profiles of the grapevine leaves, as well as baseline compositional levels before pathogen infection between those *Vitis* species with differing resistance levels.

Far beyond providing the mechanical support of living cells, the plant cell wall acts as one of the first barriers in the frontline of plant–pathogen interactions [22]. It is widely recognized that the plant cell wall is among the most complex structures in nature, generally thought to consist of cellulose microfibrils embedded in a pectin and hemicellulose matrix [23]. However, the composition and conformation of these polysaccharide polymers are found to vary among the species, varieties and tissue types, as well as the developmental stages sampled [24]. These differences are necessary to ensure the functional processes linked to biological activities are performed effectively (e.g., fruit ripening, stem elongation and pollen tube formation) [24]. Furthermore, a number of previous research studies have indicated that some polysaccharide polymers can increase the rigidity and strength of the cell wall for protection [25,26,27]. However, the work on profiling technologies for the cell wall at the polysaccharide level was a challenge for many decades due to the limitations of the analytical techniques available. The recent development of carbohydrate microarray technology using monoclonal antibodies has overcome these issues and is able to generate high-resolution profiles of polysaccharide epitopes. This provides information-rich datasets for comparative studies on cell wall changes and new insights to be gained [28]. To date, this method has successfully been applied to a number of grapevine studies, mainly analyzing grape berries [29,30,31,32] and one on the study of leaves (cultivar “Shiraz”) [33]. These studies provide a valuable foundation, which can now be extended to pathogen infection research.

In this study, we applied carbohydrate microarray technology combined with other analytical cell wall methods to profile grapevine leaves for possible associations between cell wall polymer composition and grapevine downy mildew resistance. The Chinese native species (*V. amurensis* cv. “Shuanghong”) with higher downy mildew resistance and European species (*V. vinifera* cv. “Cabernet Sauvignon”) with lower resistance were chosen for leaf sampling. Furthermore, one hybrid (crossed from these two species) was also profiled for comparison as it showed partial resistance.

## 2. Material and Methods

### 2.1. Plant Materials and Disease Resistance Assessment

*Vitis vinifera* cv. “Cabernet Sauvignon” (susceptible, hereafter shortened as “Cab”), *Vitis amurensis* cv. Shuanghong (resistant, hereafter shortened as “SH”) and their hybrid (partially resistant, hereafter shortened as “Cross”) were maintained in the germplasm vineyard in the Center for Viticulture and Enology, Shanghai Jiao Tong University, Minhang District, Shanghai, China. To verify the resistance of selected cultivars, the fully expanded leaves (4th leaf from tip) were picked, and the leaf disks (9 discs each, approx. 11 mm in diameter) were generated using a cork borer. The leaf discs were then incubated in a Petri dish with moist filter paper underneath. The downy mildew broth (35-µL, 10^5^ sporangia/mL) was inoculated into the center of the leaf discs, and the Petri dishes were incubated in a culture chamber at 22 °C with a photoperiod of 16/8 h (light/dark, respectively) for 7 days. The infection was investigated using microscopy, and the severity index was calculated according to Yu et al. [15].

### 2.2. Cell Wall Preparation

The grapevine leaves (3 biological replicates) were processed to alcohol insoluble residue (AIR) [31] for later cell wall analysis. Briefly, after storage at −80 °C overnight, leaf samples were ground with liquid nitrogen using a high-throughput Tissue-Lyser (JXFSTRP series, Shanghai Jingxin Industrial Development Co. Ltd., Shanghai, China) for 30 s at 50 Hz. In order to deactivate any endogenous enzymes, four volumes of 70% ethanol were added to the homogenized samples and then incubated at 95 °C for 15 min using a water bath. The insoluble residue was recovered by centrifugation and then sequentially washed with Methanol, Chloroform, Acetone (2 h each and the pelleted residue retained after centrifugation). Finally, the washed residue was dried under a fume-hood and then re-suspended with the same volume of dH_2_O, stored at −80 °C, before being freeze-dried to form AIR powder.

### 2.3. Comprehensive Microarray Polymer Profiling (CoMPP)

CoMPP was performed on the AIR sourced from the grapevine leaves [28]. Briefly, 10 mg of AIR was firstly extracted with CDTA (diamino-cyclohexane-tetraacetic acid, 50 mM, pH 7.5), and later, an alkali extraction using NaOH (4 M + 0.1% NaBH_4_) was performed. These two solvents were routinely used for obtaining the pectin-rich fraction (CDTA) and the hemicellulose fraction (NaOH) [28,29,30,31]. Each extraction was carried out in a 300 µL solution, and the tubes (each containing a glass bead) were agitated at a frequency of 30 Hz for 2 min followed by 6 Hz for 2 h. This was performed sequentially for each extract, so a total of 300 µL CDTA extract and 300 µL NaOH extract was obtained from 10 mg AIR. Each sample was prepared as 4 dilutions (first: 1:1, followed by serial five-fold dilutions of the preceding sample) before they were pipetted into 384-microwell plates. All of the samples (including dilutions) were then printed onto nitrocellulose membranes (pore size 0.45 mm, Whatman) using a microarray instrument (Marathon, Arrayjet, Roslin, UK). The printed arrays were firstly blocked with phosphate buffered saline (140 mM NaCl, 10 mM Na_2_HPO_4_, 2,7 mM KCl, 1.7 mM KH_2_PO_4_, PH 7.5) with 5% (*w*/*v*) low-fat milk powder for 1 h followed by an overnight incubation at 5 °C with selected polysaccharide monoclonal antibodies (mAbs) or carbohydrate binding modules (CBMs) (see Table 1). The secondary binding was later performed in PBS +5% Milk powder for 2 h with anti-rat, anti-mouse, or anti-His tag antibodies conjugated with alkaline phosphatase (Sigma) diluted 1:5000 (anti-mouse and anti-rat) or 1:1500 (anti-His tag). Arrays were washed in PBS and then developed in a chromogenic solution (5-bromo-4-chloro-3-indolylphaphate and nitro blue tetrazolium in alkaline phosphatase buffer (100 mM NaCl, 100 mM diethanolamine, 5 mM MgCl_2_, pH 9.5). The arrays were dried and scanned by a Cano Scan 8800 F scanner (SØborg, Denmark) with a signal intensity at 2400 dots per inch (dpi) and saved as a TIFF image. The probed arrays were quantified by using Array-ProAnalyzer 6.3 (Media Cybernetics, Rockville, MD, USA) software. The data were presented in a heatmap format, as the relative percentage distribution between the samples and antibodies, where the maximal mean spt signal intensity was set to 100%, and the rest of values were normalized accordingly. A cut-off of 5% was applied to limit the signal noise.

### 2.4. Gas Chromatography–Mass Spectrometry (GC–MS) for Monosaccharides

The monosaccharide composition of grapevine leaf AIR was determined by using GC–MS [31]. Briefly, 5 mg of AIR was hydrolyzed with 2 M Trifluoroacetic acid (TFA) at 110 °C for 2 h in the screwed tubes. The tubes were then cooled and centrifuged to collect the supernatant. The soluble monosaccharides in the supernatant were converted to methoxy sugars using 1 M methanolic HCl at 80 °C for 16 h. The methoxy sugars were then used to perform the silylation by adding a pre-mixed solution (HMDS + TMCS + Pyridine, 3:1:9, Sylon HTP kit; Sigma-Aldrich, MO, USA) and then incubated at 80 °C for 20 min. The TMS glycosides were dissolved in cyclohexane and injected onto a 7890 B (Agilent, Palo Alto, CA, USA) gas chromatograph coupled to an Agilent 5877 MS Mass spectrometer. The GC/MS was performed at the Instrumental Analysis Center, Shanghai Jiao Tong University. A sugar mixture that contained 7 main monosaccharides (arabinose, rhamnose, xylose, galactose, galacturonic acid, mannose, glucose) was diluted to five concentrations, and underwent the same process (silylation and GC–MS) for creating the standard curves. The main monosaccharides from leaf AIR were quantified using established standard curves and myo-inositol as the internal standard.

### 2.5. Fourier Transform Infrared (FTIR) Spectroscopy

AIR sourced from grapevine leaves was scanned using Nicolet 6700 Fourier Transform Infrared Spectroscopy (Thermo Scientific, MA, USA) containing a Golden Gate Diamond Attenuated Total Reflectance (ATR) accessory with a type II diamond crystal; the spectra between 4000 and 650 cm^−1^ were recorded with a Geon-KBr beam splitter and DTGS/Csl detector. Each measurement consisted of 128 co-added scans, and the spectral data were processed using OMNIC software.

### 2.6. Univariate Statistical and Multivariate Data Analysis

All samples were analyzed for statistical significance by using Microsoft Excel 2020, and Univariate statistical analyses were performed (ANOVA, with *p* = 0.05). Multivariate analysis for CoMPP raw data was performed with the SIMCA 14 software package (Sartorium Stedim Biotech—Umetrics AB, Umea, Sweden).

## 3. Results and Discussion

### 3.1. Downy Mildew Resistance Measurement of Leaf Disks Sourced from the Two Grapevine Cultivars and the Hybrid Cross

Leaves from cultivar “SH”, “Cab” and “Cross” were subjected to downy mildew resistance measurement before implementing cell wall analysis. The inoculation of *Plasmopara viticola* onto the three cultivars is shown in Figure 1. The visual observation clearly indicated that cultivar “SH’ showed high resistance to mildew and cultivar “Cab” showed the most susceptibility, whereas the cultivar “Cross” (bred from “SH” and “Cab”) showed partial resistance. The disease severity index was calculated according to Yu et al. [15] and showed similar trends by visual observation (data not shown); these results generally confirmed the previous finding [15] and provide validation of the chosen samples for later cell wall analysis.

### 3.2. Comprehensive Microarray Polymer Profiling (CoMPP)

Comprehensive Microarray Polymer Profiling uses a set of mAbs and CBMs [28] to recognize the structural motifs of polysaccharides and glycoproteins present in extracted fractions from cell wall AIR samples. CoMPP generates datasets on the polysaccharide epitope occurrences rather than their monosaccharide composition, and, therefore, provides in-depth information on the cell wall epitope structure based on antibody characterisation. In this study, two sequentially extracted fractions (CDTA and NaOH) from the grapevine leaf cell wall AIR were used for analysis, and the datasets obtained were used for constructing the heatmaps and multivariate data analysis. The heatmap generated from the CDTA fraction of cultivar “Cab” was found to be rich in pectin, specifically in homogalacturonans (HGs) and rhamnogalacturonan I-associated polymers (Figure 2). The HGs were recognized by mAbs JIM5, JIM7, LM18, LM19 and LM20. The selection of various mAbs for HGs depends on their affinities to the pectin and the corresponding specific structures and conformations (Table 1). Another two mAbs (LM7 for partially methyl-esterified epitope of HG that results from non-blockwise de-esterification processes, and LM8 for xylogalacturonan) were also used in the analysis; however, a very weak signal was detected, and their relative abundance were recorded as zero in the heatmap due to the cut-off (<5) imposed. The mAbs INRA-RU1 and RU2 were used for probing the grapevine leaf cell wall for the first time in this study, as the first CoMPP analysis on grapevine leaf was performed on the cultivar “Shiraz” without using any INRA RG I backbone mAbs [33]. The leaf cell wall profile shows relatively lower epitope abundance on RG I than HG compared to grape berry cell walls [31,32,34]; this indicates the tissue-specific feature of plant cell wall variation. Interestingly, INRA-RU2 shows much lower abundance than INRA-RU1. A previous specificity study showed that INRA-RU1 recognizes RG I with more than six disaccharide backbone repeats, but the affinity decreases steeply as the number of disaccharides increases beyond seven, whereas INRA-RU2 requires at least two disaccharides with the highest affinity to seven disaccharides [35]. The differences between the above two mAbs may be due to the higher avidity of INRA-RU1 to RGI, especially the acetylesterified RGI decorated with short galactan side chains [35]. In addition to the RG I main chain, the structural motif abundance of mAbs LM5 and LM6 also revealed the presence of galactan and arabinan structural motifs, consistent with branched pectin RG I domains. Apart from the pectin components, xyloglucan (recognized by mAbs LM15 and LM25) was also found in the CDTA fraction with a relatively low signal. This phenomenon was also found in a number of previous studies, which suggested the association between pectin and xyloglucan [4,28,36,37]. Furthermore, the structural motifs of glycoproteins have also been detected by mAbs JIM8 and JIM13; other mAbs for AGP showed relatively lower signals and then were cut off from heatmap (Figure 2). The CoMPP CDTA heatmap of “Cab” shows generally similar profiles to “Shiraz” [33] with the presence of some differences. In Shiraz leaf cell walls, relatively higher epitope abundance was detected on mAbs JIM7 and LM20, which recognize highly esterified HG; this may indicate that the esterification level is higher in Shiraz leaf pectin than that in Cabernet Sauvignon.

In order to investigate the possible associations between cell wall polymers and grapevine downy mildew resistance, the cultivars with high resistance (“SH”: *Vitis amurensis* cv. Shuanghong) and mid-resistance (“Cross”: hybrid from Shuanghong and Cabernet Sauvignon) were also used for CoMPP analysis. In general, the heatmap (Figure 2) also shows the presence of HG, RG I, galactan, arabinan, xyloglucan and AGP epitopes in the CDTA fraction; however, a number of differences are noted compared to cultivar “Cab”. Firstly, cultivar “SH” showed a generally higher epitope abundance on all the mAbs which recognize HG, especially those two mAbs (JIM7 and LM20) with higher affinity to methylesterifed HG. Interestingly, cultivar “Cross” showed a markedly higher abundance of mAbs JIM7 and LM20 than samples “Cab” but lower than samples “SH”. In the plant cell, pectin is primarily synthesized in highly methyl-esterified form from the Golgi apparatus and then de-esterified with the action of endogenous pectin methyl esterases (PMEs); on the other hand, the plant could also secrete inhibitory proteins (PMEi) which block the activity of PME to precisely regulate the esterification level of the pectin during development. The status of pectin could also influence the porosity and integrity of the cell wall [38,39]. The pectin main chain with can be highly methylated will need the additional enzymes for thorough breakdown, and, therefore, decrease the availability of substrates targeted by other pathogenic cell wall-degrading enzymes (CWDEs), such as hemicellulases and cellulases [40]. Previous studies on some crops (tomato, cotton and wheat) have suggested that a high degree of esterification (DE) level of pectin is positively correlated with plant defense. [4,41,42]. In addition to differences in HG, cultivar “SH” showed the highest abundance for RG I and its associated side chains (arabinan and galactan) than the other two cultivars. RG I with side chains is usually called the hairy region of pectin; its complete deconstruction requires a set of lytic enzymes, such as RG lyases, arabinanases and galactanases. Previous studies have suggested the roles of arabinan and galactan side chains as cross-linked bridges between pectin and xyloglucan [23,43,44,45]. In addition to structural support, these side chains have been suggested to be involved in other biological activities, e.g., galactans could be used as structural markers, as their decreasing signal is found during fruit ripening [30]. Arabinan has been focused on for its potential roles in the stress tolerance of plants, such as in regard to drought [46] and salt stress [47]. However, currently no direct connection between arabinans and pathogen resistance has been confirmed. Apart from the pectin changes, the heatmap does not show notable differences in the epitope abundances of mAbs recognizing hemicelluloses (e.g., LM15 and LM25), even though some minor variations are present between the replicates of each of the cultivars. The comparison of glycoprotein profiles revealed the marked differences between three cultivars, and “SH” showed the highest abundance on the mAb JIM8 (which recognizes an AGP epitope). Recent reviews have suggested additional roles for AGPs in plant stress tolerance compared with more well-established roles in plant growth and development; AGP accumulation has been observed during a range of biotic and abiotic stresses [47,48,49].

The CoMPP heatmap gives a general overview of the polysaccharide profile for initial interpretation; however, it compresses the data for obtaining the relative values, and numerous variables (mAb abundance) can lead to a greater difficulty in visually inspecting datasets for differences. Thus, multivariate data analysis was routinely applied to the raw data to provide additional information (Figure 3). SIMCA software was used in this study to implement an unsupervised method (principal component analysis, PCA), in order to obtain an unbiased overview of the data structure. The raw CoMPP dataset was pre-processed in Microsoft Excel, and the mAbs were set as variables. The pre-processed dataset was then transferred to SIMCA to construct the PCA plot. The algorithms, if PCA found the maximum variance in the data assigning it to principal component 1 (PC1), then proceeded to PC2, PC3, etc. In this study, the PCA score plot, which contained two components (PC1 and PC2) with high quality (Figure 3A), was constructed; the score plot shows that three cultivars were clustered into three groups, and cultivar “SH” was separated from others (PC1 = 41.6%). The loading plot (Figure 3B) shows that the variables (mAbs) that drive this separation mainly include HG, RG I with associated side chains and AGPs. Interestingly, two replicates of cultivar “Cross” sit at the same side as cultivar “SH 1” and “SH 2” and the major variables driving this separation in the vertical dimension (PC2 = 26.3%) are mAbs LM18 and LM19, which recognize the partially esterified HG. Further group-to-group (SH to Cross) comparisons suggest that corresponding variables include JIM7 and LM 20, which recognize methylesterifed HG (data not shown); this again supports the heatmap datasets. In addition to pectin, mAb BS-400-2 (recognizes callose), which did not show zero on heatmap, however, shows its association with other pectin recognizing mAbs in the PCA plot. Callose has been well documented in many studies for its roles in plant/microbial interactions; it has been found to be deposited around the infection sites to block the pathogens [15,40].

The CoMPP datasets of the NaOH fraction of the three cultivars yielded a heatmap (Figure 4) rich in hemicellulosic polymers, including mannans (mAb LM21), xyloglucan (mAbs LM15, LM24 and LM25) and xylan (mAb LM11). Glucan, cellulose and AGP were also detected by mAbs (mAbs BS-400-2, CBM3a, JIM8, JIM13), as well as the signals of galactan (mAb LM5) and arabinan (mAbs LM6 and LM13); these may again support the hypothesis that these side chains act as the linkages between RG I and hemicellulosic polymers [45]. The comparison between cultivars revealed the increasing trend of xyloglucan (with XXXG motif, recognized by mAb LM15). Hemicellulose polymers have been suggested to play critical roles in the strengthening of the plant cell wall structure, therefore elevating the plant resistance to stress. For example, the marked increase in arabinoxyloglucan has been stated to positively correlated with the resistance of tobacco to *Botrytis cinerea* [50]; the silencing of xylanases activity increased the xylan content and silique dehiscence resistance [51]; and a generally enriched hemicellulose was found in resistant wheat cultivar compared to cultivar susceptible to Fusarium Head Blight [52]. In addition to mAbs LM15, another hemicellulosic polymer, (galacto)(gluco)mannan (mAb LM21), showed a similar increasing pattern. Interestingly, similar to the CDTA fraction (Figure 2), an increasing trend was also observed for AGP epitope abundance, recognized by both mAbs JIM8 and JIM13, which again suggested their possible roles in grapevine/downy mildew interactions. To obtain further information on possible associations between polymers, the PCA score plot (Figure 5A) and loading plot (Figure 5B) were constructed from the CoMPP raw datasets of the NaOH fraction. In Figure 5A, it is very clear that those samples were well clustered into three groups on the horizontal dimension; cultivar “SH” has distinct distance from cultivar “Cab”, whereas cultivar “Cross” sits between them (PC1 explained by 28.6% variance). In the loading plot (Figure 4), the main variables driving this separation include AGP, mannan, xyloglucan and cellulose, which again support the heatmap datasets (Figure 4).

### 3.3. Monosaccharide Composition of Leaf Cell Wall Materials Using GC–MS

Gas chromatography (GC) has been routinely applied in cell wall analysis; it gives additional monosaccharide composition data for the validation of CoMPP analysis [32,33,37]. In this study, GC was applied to the most distinct samples (”SH” and “Cab”), and seven monosaccharides were analyzed (Figure 6). In general, leaf AIR from two cultivars contains a substantial amount of GalA (ca. 35–40 mol.%), with the presence of Ara (ca. 14–15%), Rha (ca. 5–7%) and Gal (ca. 13–14%), which suggests that leaf cell walls contain homogalacturonan (HG), rhamnogalacturonan I (RG I) and possibly rhamnogalacturonan II (RG II). On the other hand, the presence of Xyl (ca. 11–12%), Man (ca. 4–6%) and Glc (ca. 12–14%) suggests that xylan, xyloglucan and mannan are major components of the hemicellulosic fraction of grapevine leaf cell walls. These datasets show high similarity to the leaf sample of cultivar “Shiraz” [33], but different to the composition of tobacco leaves [36] and grape berry cell walls [37], this indicates the cell wall differences among the plant organs and tissues. Furthermore, the comparison between “Cab” and “SH” did not reveal significant differences among those monosaccharides; In addition to the markedly higher mol.% of GalA in cultivar “SH”, this may explain the generally higher abundance of mAbs that recognize HG (Figure 2). This result highlights the limitations of GC composition datasets, which tend not provide information on subtle structural differences, whereas CoMPP provides additional richer and complementary data on the polysaccharide structure by virtue of epitope abundance. In a previous study on grape berry cell wall deconstruction, the GC–MS did not show any significant changes between samples, whereas CoMPP was able to detect changes such as pectin de-esterification [37].

### 3.4. IR Spectroscopy 

ATR-FT-MIR spectroscopy provides a fast and convenient method to detect the chemical functional groups present in cell wall samples [53]; it is usually applied in cell wall studies to provide a general overview of vibrational chemical bonds linked to polysaccharides and protein [31,33]. In this study, AIR samples sourced from cultivar “SH” and “Cab” were used for IR spectroscopy, generating wavescan spectral profiles (Figure 7) which can be used for comparisons. Two spectra were generated, both consisting of a number of maxima, including pectin HG at 1017 cm^−1^, xyloglucan at 1041 cm^−1^, amides at 1650 cm^−1^ and esterified pectin at 1740 cm^−1^ [53]. The comparison showed a similar spectral profile in general, which confirmed the finding from GC–MS and CoMPP, and reflects that bulk chemistry does not vary dramatically. However, few differences are worth noting: First, visual inspection indicated subtle difference in the shape of maxima at 1041 cm^−1^, which may suggest the structural differences in hemicellulosic polymers; this may be explained by the higher abundance of xyloglucan (with XXXG motif) found in Figure 4. The difference in maxima at 1017 cm^−1^ was also noted, which may indicate the higher pectin HG content in the “SH” AIR sample; however, the supporting data will be needed in future work as the GC–MS and CoMPP mainly focused on the non-cellulose fractions. Interestingly, the most significant difference between the two spectra is the maxima (1740 cm^−1^), which refer to esterified HG. This again supports the finding from Figure 2 and Figure 3 that indicated the cultivar “SH” has a higher abundance of esterified HG.

## 4. Conclusions

In order to achieve the successful penetration of plant tissue, pathogenic microorganisms (e.g., fungi) are found to employ various methods in their arsenal. Among these, the secretion of cell wall-degrading enzymes (CWDEs) is recognized as an effective and precise way for unravelling and degrading one of the most complex structures in the nature. Compared to rot fungi in general, which secretes a wide range of CWDEs, as is the case for necrotrophs, for thorough degradation, the bioinformatic genome and transcriptome comparisons suggest that the relatively lower diversity of lytic enzymes is expressed in the oomycete family [54]. Thus, the interactions between plants and oomycetes in the cell wall battlefield may vary during fungal infection. However, there is very limited information on the host resistance from the viewpoint of the plant cell wall structure. Previous studies found that grape hybrids which were bred from native species may acquire parental resistance to differing degrees. In this study, cutting-edge cell wall profiling tools were applied to selected grapevine leaf samples to investigate the possible correlations between grape leaf cell wall polymers and downy mildew resistance.

The carbohydrate microarray provided much richer information than more classical methods (GC–MS and FTIR), and revealed a number of differences between the uninfected leaves of *Vitis amurensis* cv. “Shuanghong” (resistance) and *Vitis vinifera* cv. “Cabernet Sauvignon” (susceptible), with the aid of multivariate data analysis tools. Significantly higher amounts of methyl esterified HG were found in cultivar “SH”, and the increasing trend of this component from cultivar “Cab” and “Cross” (hybrid with moderate resistance) to “SH” may suggest its potential role as a structural biomarker in downy mildew resistance; this was also supported by FTIR spectral data of the functional chemical groups. Similar patterns were also noted for xyloglucan with the XXXG motif. Arabinogalactan proteins (AGPs) were found to be the highest in abundance in resistant “SH” but did not show significant differences between “Cab” and “Cross”.

In summary, the higher esterification level of HG, high content of mannan and xyloglucan (XXXG) could be the correlated with the native Chinese grapevine (*Vitis amurensis* cv. “SH”) natural resistance to downy mildew, and may be inherited from the native grapevine species through breeding. However, are these differences common to other wild grapevine species with similar resistances? How are these polymers regulated in wild grapevines during development? In order to answer these questions, the relevant genes behind these cell wall polymer changes will need to be focused on, e.g., the balance in expression of endogenous pectin methyl esterase (PME) and PME-inhibiting proteins (PMEI). Furthermore, in addition to the structural enhancement of the cell wall ahead of infection, we do not know how these wall polymers are altered by mildew infection/invasion. Future work will be needed to observe the induced changes of these cell wall components during mildew infection, as well as relevant factors (e.g., lytic enzymes), for obtaining an in-depth understanding of the complex network involved in cell wall regulation and molecular architecture. This is the first comprehensive comparison study of leaf cell wall polysaccharide polymer/protein variation from Vitis species and hybrids differing in their resistance to grapevine downy mildew. This provides important baseline reference datasets for future investigations on other biotic and abiotic stresses important in agricultural grape production.

## Figures and Tables

**Figure 1 polymers-13-01379-f001:**
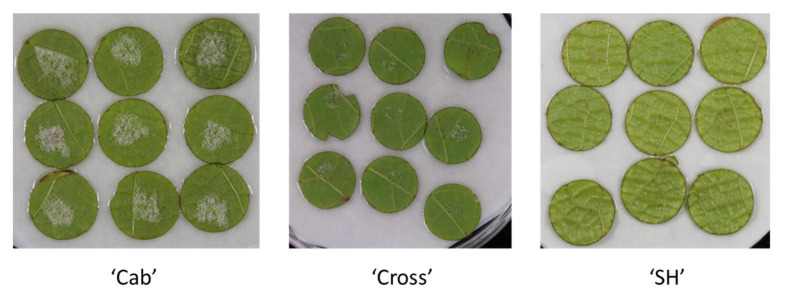
Grapevine downy mildew symptoms on leaf disks with different sporulation density observed at 7 days post-inoculation. “Cab”: *Vitis vinifera* cv. “Cabernet sauvignon”; “SH”: *Vitis amurensis* cv. “Shuanghong”; “Cross”: the hybrid from “Cab” and “SH”.

**Figure 2 polymers-13-01379-f002:**
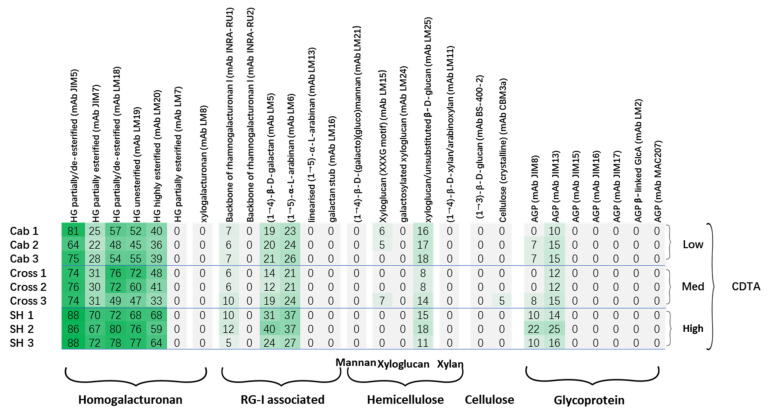
Comprehensive Microarray Polymer Profiling (CoMPP) analysis of CDTA extract of grapevine leaf cell wall AIR samples. The heatmap shows the relative abundance of plant cell wall glycan-associated structural motifs. The highest signal was set as 100, and others were adjusted accordingly; the color intensity is correlated to the mean spot signal. A cut-off (<5) was applied.

**Figure 3 polymers-13-01379-f003:**
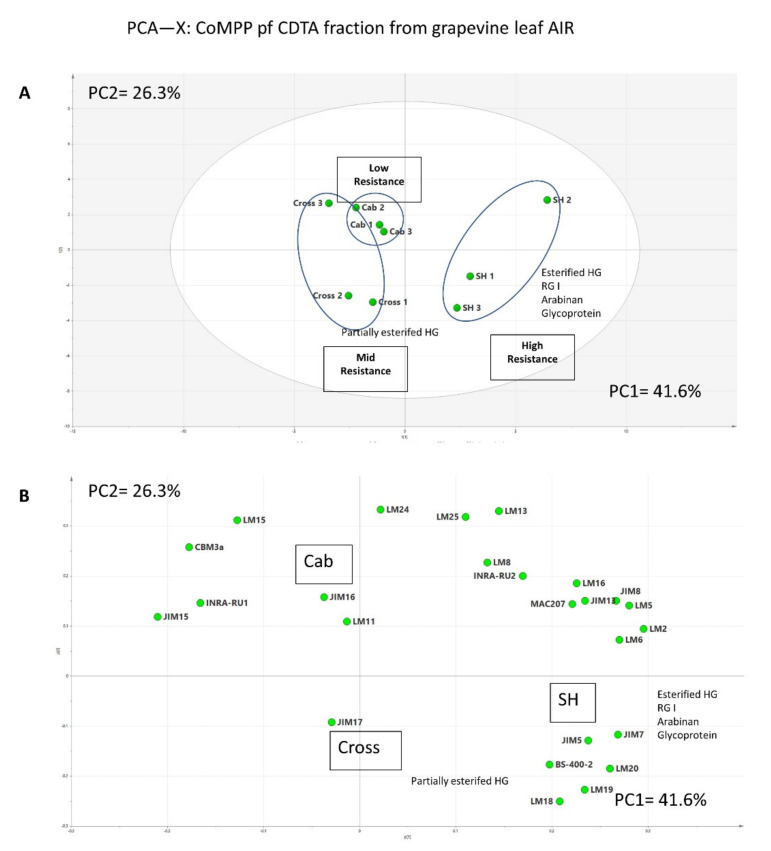
PCA score plot (**A**) and loading plot (**B**) of the CDTA (pectin-rich) extract from AIR sourced from grapevine leaves.

**Figure 4 polymers-13-01379-f004:**
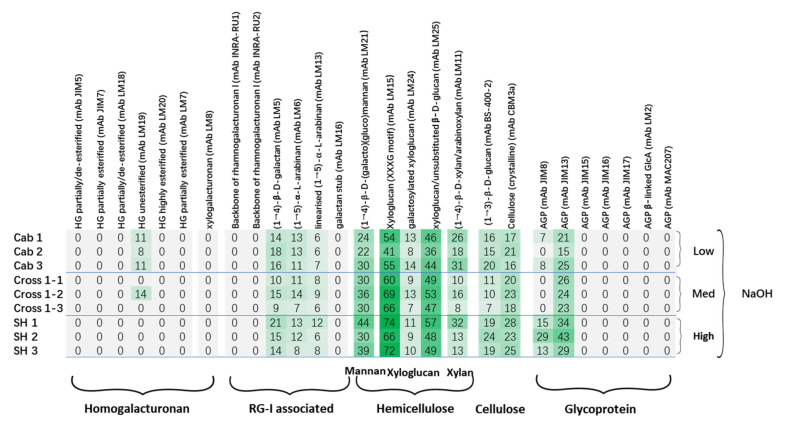
Comprehensive Microarray Polymer Profiling (CoMPP) analysis of NaOH extract of grapevine leaf cell wall AIR samples. The heatmap showed the relative abundance of plant cell wall glycan-associated structural motifs. The highest signal was set as 100, and others were adjusted accordingly; the color intensity is correlated to the mean spot signal. A cut-off (<5) was applied.

**Figure 5 polymers-13-01379-f005:**
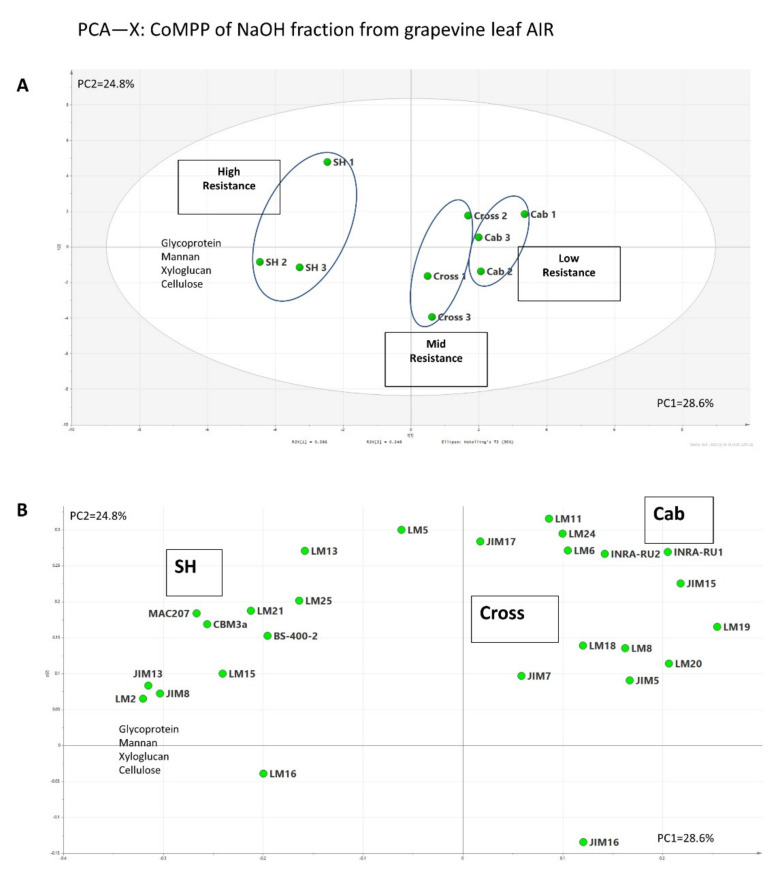
PCA score plot (**A**) and loading plot (**B**) of the NaOH (hemicellulose-rich) extract from AIR sourced from grapevine leaves.

**Figure 6 polymers-13-01379-f006:**
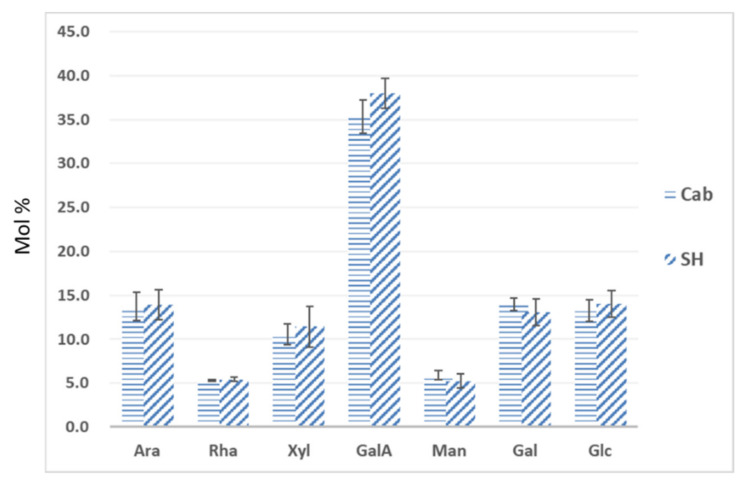
The monosaccharide composition of AIR sourced from grapevine leaves. The composition is expressed in relative mol% of seven main monosaccharides present in grapevine leaf cell wall. Ara: arabinose; Rha: rhamnose; Xyl: xylose; GalA: galacturonic acid; Man: mannose; Gal: galactose; Glc: glucose. Error bars represent the standard derivation of the mean value of three biological repeat.

**Figure 7 polymers-13-01379-f007:**
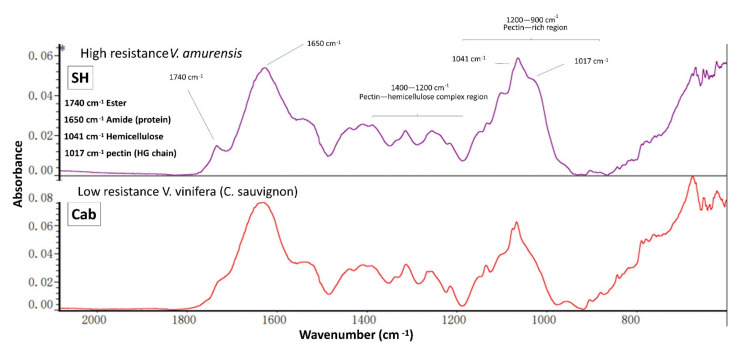
Fourier transform infrared (FT-IR) spectra generated from the AIR samples sourced from grapevine leaves.

**Table 1 polymers-13-01379-t001:** mAbs and CBMs used in the CoMPP analysis. HG: homogalacturonan; RG I: Rhamnogalacturonan I.

Code	Specificity	Group
**JIM5**	Binds to partially methyl esterified HG	**HG**
**JIM7**	Binds to heavily methyl esterified HG	
**LM18**	Binds to de-esterified HG, higher affinity to shorter chain (DP < 4)	
**LM19**	Binds to de-esterified HG, higher affinity to longer chain (DP > 4)	
**LM20**	Binds to methyl-esterified HG	
**LM8**	Binds to xylogalacturonan	
**INRA-RU1**	Binds to unbranched region of RGI, need >6 disaccharide backbone repeats, maximum binding to DP = 14	**RGI and side chains**
**INRA-RU2**	Binds to unbranched region of RGI, significant binding to 2 disaccharide backbone repeats, need at least DP = 4	
**LM5**	Binds to (1–4)-ß-D-galactan	
**LM6**	Binds to (1,5)-α-L-arabinan, may bind to AGP	
**LM13**	Binds to linear arabinan, highly sensitive to arabinanase	
**LM16**	Binds to galactan stub	
**LM21**	binds to mannans, glucomannans, and galactomannans. Binds most strongly to mannotetraose and mannopentaose.	**Hemicellulose**
**BS-400-2**	Binds to (1–3)-ß-D-glucan	
**LM15**	Binds to xyloglucan, XXXG motif	
**LM24**	Binds to XLLG or XXLG	
**LM25**	Binds to xyloglucan/unsubstituted glucan,	
**LM10**	(1–4)-ß-D-xylan	
**LM11**	bind to unsubstituted xylans and arabinoxylans carrying a low degree of arabinose substitution	
**CBM3a**	Binds to crystalline cellulose and can detect cellulose in both in vitro assays and directly in plant materials	**Cellulose**
**JIM8**	Arabinogalactan protein	**Glycoprotein**
**JIM13**	Arabinogalactan protein	
**JIM15**	Arabinogalactan protein	
**JIM16**	Arabinogalactan protein	
**JIM17**	Arabinogalactan protein	
**LM2**	Arabinogalactan protein	
**MAC207**	Arabinogalactan protein	

## Data Availability

Not applicable.

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
