# Peer review of "Comprehensive Leaf Cell Wall Analysis Using Carbohydrate Microarrays Reveals Polysaccharide-Level Variation between Vitis Species with Differing Resistance to Downy Mildew"

_polymers, 2021, doi:10.3390/polym13091379_

Round 1

Reviewer 1 Report

It seems to me that CoMMP is a very reasonable approach to investigating the role of cell walls in disease resistance, however the results were not as consistent as I would hope for.

I’m afraid the authors have been a little sloppy in their proofreading of the manuscript. E.g. on line 114 it reads that there were 105 sporangia per ml, but in a previous paper, ref 15, they had 105 sporangia per ml! They have the title of ref 15 a “The model of host resistance….” Whereas it is really “The mode of host resistance…”

The paper could use a lot of help with the English. In many places they don’t use standard grammar, and also they use inappropriate words frequently.

The authors are mistaken on line 224. The INRA- RU2 mAb has highest affinity to 6,7,or 8 repeats. It shows some, but much less to the 2 repeat oligosaccharide.

I’m afraid I do not have a clear understanding of PCA, but I think I have the gist of it. So, I have many questions. What raw data was used as the data set? You only show three replicates in the heat maps and some of the spots showed no response in all lines, so the variance should be zero, yet they are there in the loading plots. Did you use results from all of the dilutions? If so  were they corrected for the dilution factor? I worry that there was too much variation between replicates. Is that normal for the CoMPP approach? What does the PCA add to the results? We can see from the heat maps that methylesterification of pectins is correlated with resistance. It seems to me that PCA makes the results less clear by combining many features at once. It  might be useful for predicting whether a particular cross might be resistant by looking at where its principal components cluster.

Author Response

Dear Editors

  We would like to express our appreciation to you and corresponding reviewers for the peer review and generous comments on the submitted paper Manuscript ID: polymers-1145339 Comprehensive leaf cell wall analysis using carbohydrate microarrays reveals polysaccharide-level variation between Vitis species with differing resistance to downy mildew”. We have edited the manuscript by using the track change function (for improving the English) in word and highlighted (in yellow) the corrected special points which the reviewers addressed. Please see below for the responses (in blue). The reviews and our revision accordingly helped us to improve the manuscript.

Reviewer 1.

It seems to me that CoMMP is a very reasonable approach to investigating the role of cell walls in disease resistance, however the results were not as consistent as I would hope for.

Response: We thank the reviewer for the positive assessment of this approach. CoMPP has been applied to a number of food processing research systems and provide datasets with very good consistency, this may due to the similarity of starting materials (e.g. grape berry at same ripening stages). However, in the analysis of leaf cell walls, it is more difficult to achieve the same level of consistency as the leaf is perhaps faster growing, developing stage, the biological repeats came from 3 vines and we picked the samples according to their size and position on the shoot. We have tried to minimize the variance between leaves but we can imagine the differences/variability are always present. However, by comparing the cultivars with distinct characteristics (e.g. disease resistance in this study), trends could be observed that support the differences correlated with resistance as being significant.   

I’m afraid the authors have been a little sloppy in their proofreading of the manuscript. E.g. on line 114 it reads that there were 105 sporangia per ml, but in a previous paper, ref 15, they had 105 sporangia per ml! They have the title of ref 15 a “The model of host resistance….” Whereas it is really “The mode of host resistance…”

Response: We thank the reviewer to point this out. We are apologized about those mistakes and have corrected them, we have also checked the manuscript thoroughly to avoid other similar mistakes.

The paper could use a lot of help with the English. In many places they don’t use standard grammar, and also they use inappropriate words frequently.

Response: We thank the reviewer in pointing this out, we have edited this manuscript thoroughly to improve the English.

The authors are mistaken on line 224. The INRA- RU2 mAb has highest affinity to 6,7,or 8 repeats. It shows some, but much less to the 2 repeat oligosaccharide.

Response: We much appreciate to the reviewer for pointing this out, we have corrected this and also added more information on the differences between the two mAbs. I believe this information helps clarify the results obtained.

I’m afraid I do not have a clear understanding of PCA, but I think I have the gist of it. So, I have many questions. What raw data was used as the data set? You only show three replicates in the heat maps and some of the spots showed no response in all lines, so the variance should be zero, yet they are there in the loading plots. Did you use results from all of the dilutions? If so were they corrected for the dilution factor? I worry that there was too much variation between replicates. Is that normal for the CoMPP approach?

Response: We thank the reviewer for pointing this out as other readers may also have those questions. In the analysis of CoMPP, the raw datasets (signal intensity quantified by CanoScan instrument) are usually processed by Microsoft Excel to create the heatmap for rapid interpretation and comparison. In the heatmap, the highest signal intensity is usually set as 100 and others are adjusted accordingly, the adjusted value which below 5 usually will be cut-off (shown as zero) for easier interpretation. However, those adjustment compress the data and we may lose some information, so we usually construct the PCA plot by using raw dataset to give additional support and information, as you noticed, these variances (shown as zero) are again present in loading plot. So far, this approach is standard in CoMPP analysis and has provided many important information on berry ripening and winemaking studies, we have also published this approach on the recently published protocol book (please see below)

Moore, J.P., Gao, Y., et al, 2020. Analysis of Plant Cell Walls Using High-Throughput Profiling Techniques with Multivariate Methods. Zoe¨ A. Popper (ed.), The Plant Cell Wall: Methods and Protocols, Methods in Molecular Biology, vol. 2149,

https://doi.org/10.1007/978-1-0716-0621-6_18, © Springer Science+Business Media, LLC, part of Springer Nature 2020  

Thanks again for the reviewer to point this out, we have added some more information on the data processing to make readers more easily understand.

What does the PCA add to the results? We can see from the heat maps that methylesterification of pectins is correlated with resistance. It seems to me that PCA makes the results less clear by combining many features at once. It might be useful for predicting whether a particular cross might be resistant by looking at where its principal components cluster.

Response: As we discussed in above question. Previous studies have suggested that PCA could provide additional information to the heatmap, as it not only clusters those samples into different classes, but also help us to indicate the possible correlations between cell wall polymers. In this study, for example, the PCA sourced from NaOH fraction has indicated the possible correlation of β-1,3-glucan (Callose) with the high resistance, as well as the possible structural association between galactan (LM5 and LM16) and hemicellulose (LM15 and LM25). These information could give us clues for in-depth analysis of cell wall structures in future studies.   

Reviewer 2 Report

Physiologically, cell wall of leaves is a first barrier of plants that plays vital role in the interactions with the pathogens. The structural features of the cell wall polysaccharide building blocks are vital for preservation of its integrity and resistance to microbial infections. The main topic of the presented research is significant from the economical and agricultural perspective of grapevine production. In the presented manuscript the Authors pursue a comparative study of the Vitis vinifera cutivars differing in their susceptibility to infection and how it correlates with the structural features inferred from the microarray data. Although this approach does not provide the structural details of the investigated cell wall – derived polysaccharide fractions, it still identifies certain correlations reflecting the presence of resistance-related structural motifs among them. The manuscript is properly structured, however the combined Results & Discussion section makes it a bit difficult to distill the actual results from those refered to and discussed. The methods and techniques used are up-to-date and relevant for this type of studies. However, the manuscript is not free from errors and some claims lack precision – I have pointed these out in my specific comments.

Last but not least, in my opinion the language needs special attention as in the present form it obscures the overall message of the manuscript.

Specific comments:

1) Throughout the paper the Authors use the term “epitope”, “carbohydrate epitope” and “epitope on all the mAbs […] “ (page 7). I advise caution in using the term “epitope” in these contexts as it refers to a segment or structural motif in the complex glycan structure recognized by the specific monoclonal antibody. The antibodies serve as probes to identify structural motifs and in my view the “structural motif” would be a more accurate descriptor.

2) In the Materials& Methods:

  • Section 2.3 (Comprehensive Microarray Polymer Profiling) - a brief justification for the used extraction methods should be included. Also, in my view the Table 1 should be anchored to this section, as it lists the mAbs and CBMs, and it does not constitute a result.
  • Section 2.4 (GC-MS) – please define “standard sugar mixture”

3) In the Results & Discussion:

  • A description of the PCA approach could be expanded, including justification for the selection of the PC1 & PC2 components only (they account for roughly 68 % of variation for CDTA fraction and 53% for NaOH fraction). Some additional info on the pre-processing of the data would also be useful.
  • The use of abbreviations should be consistent throughout the text (see: Cab vs. CS, page 11). The galacturonic acid moiety should be abbreviated as GalA (please check text -page 11 and Figure 6).

Author Response

Dear Editors

  We would like to express our appreciation to you and corresponding reviewers for the peer review and generous comments on the submitted paper Manuscript ID: polymers-1145339 Comprehensive leaf cell wall analysis using carbohydrate microarrays reveals polysaccharide-level variation between Vitis species with differing resistance to downy mildew”. We have edited the manuscript by using the track change function (for improving the English) in word and highlighted (in yellow) the corrected special points which the reviewers addressed. Please see below for the responses (in blue). The reviews and our revision accordingly helped us to improve the manuscript.

Reviewer 2:

Physiologically, cell wall of leaves is a first barrier of plants that plays vital role in the interactions with the pathogens. The structural features of the cell wall polysaccharide building blocks are vital for preservation of its integrity and resistance to microbial infections. The main topic of the presented research is significant from the economical and agricultural perspective of grapevine production.

Response: We thank the review to recognize the importance of this study.

In the presented manuscript the Authors pursue a comparative study of the Vitis vinifera cultivars differing in their susceptibility to infection and how it correlates with the structural features inferred from the microarray data. Although this approach does not provide the structural details of the investigated cell wall – derived polysaccharide fractions, it still identifies certain correlations reflecting the presence of resistance-related structural motifs among them.

Response: We thank the reviewer for the positive comments.

The manuscript is properly structured, however the combined Results & Discussion section makes it a bit difficult to distill the actual results from those refered to and discussed.

Response: We thank the reviewer for the positive comments on the structure of manuscript. We agree with the comments regarding the Results & Discussion section, and have edited this section thoroughly to make it easier to read.

The methods and techniques used are up-to-date and relevant for this type of studies. However, the manuscript is not free from errors and some claims lack precision. Last but not least, in my opinion the language needs special attention as in the present form it obscures the overall message of the manuscript.

Response: We thank the reviewer for the positive comments on Methods section. We apologize for these errors which may cause misunderstanding, we have edited the manuscript thoroughly to improve the language.

Specific comments:

1)         Throughout the paper the Authors use the term “epitope”, “carbohydrate epitope” and “epitope on all the mAbs […] “ (page 7). I advise caution in using the term “epitope” in these contexts as it refers to a segment or structural motif in the complex glycan structure recognized by the specific monoclonal antibody. The antibodies serve as probes to identify structural motifs and in my view the “structural motif” would be a more accurate descriptor.

Response: We thank the reviewer to point this out and agree with these comments. “structural motif” is more accurate than “epitope” and we have replaced them where relevant.

2) In the Materials& Methods:

Section 2.3 (Comprehensive Microarray Polymer Profiling) - a brief justification for the used extraction methods should be included.

Response: We thank the reviewer for the valuable comments, we have added a brief justification to introduce the purpose of using these extraction solvents and methods and have cited the relevant articles in this section.

Also, in my view the Table 1 should be anchored to this section, as it lists the mAbs and CBMs, and it does not constitute a result.

Response: We agree with the reviewer and have moved the Table 1 to Methods section.

Section 2.4 (GC-MS) – please define “standard sugar mixture”

Response: We have added the description of sugar mixtures and relevant protocol in this section.

3) In the Results & Discussion:

A description of the PCA approach could be expanded, including justification for the selection of the PC1 & PC2 components only (they account for roughly 68 % of variation for CDTA fraction and 53% for NaOH fraction). Some additional info on the pre-processing of the data would also be useful.

Response: We agree with the reviewer, we have expanded the PCA approach to include the methods of data pre-processing, PCA construction and so on. PC3 was not selected due to the low quality (low prediction) of the model, which could not provide any more useful information.

The use of abbreviations should be consistent throughout the text (see: Cab vs. CS, page 11). The galacturonic acid moiety should be abbreviated as GalA (please check text -page 11 and Figure 6).

Response: We thank the reviewer for pointing this out, we have corrected those abbreviations to make them more consistent.

Round 2

Reviewer 1 Report

Thank you for addressing my queries and suggestions. I think Dr Moore really helped with the language. And, you explained your results better.